# Scalable Spike Source Localization in Extracellular Recordings using Amortized Variational Inference

**Cole L. Hurwitz**
School of Informatics
University of Edinburgh, United Kingdom
cole.hurwitz@ed.ac.uk

**Kai Xu**
School of Informatics
University of Edinburgh, United Kingdom
kai.xu@ed.ac.uk

**Akash Srivastava**
MIT–IBM Watson AI Lab
Cambridge, United States
Akash.Srivastava@ibm.com

**Alessio P. Buccino**
Department of Informatics
University of Oslo, Oslo, Norway
alessiob@ifi.uio.no

**Matthias H. Hennig**
School of Informatics
University of Edinburgh, United Kingdom
m.hennig@ed.ac.uk

## Abstract

Determining the positions of neurons in an extracellular recording is useful for investigating functional properties of the underlying neural circuitry. In this work, we present a Bayesian modelling approach for localizing the source of individual spikes on high-density, microelectrode arrays. To allow for scalable inference, we implement our model as a variational autoencoder and perform amortized variational inference. We evaluate our method on both biophysically realistic simulated and real extracellular datasets, demonstrating that it is more accurate than and can improve spike sorting performance over heuristic localization methods such as center of mass.

## 1 Introduction

Extracellular recordings, which measure local potential changes due to ionic currents flowing through cell membranes, are an essential source of data in experimental and clinical neuroscience. The most prominent signals in these recordings originate from action potentials (spikes), the all or none events neurons produce in response to inputs and transmit as outputs to other neurons. Traditionally, a small number of electrodes (channels) are used to monitor spiking activity from a few neurons simultaneously. Recent progress in microfabrication now allows for extracellular recordings from thousands of neurons using microelectrode arrays (MEAs), which have thousands of closely spaced electrodes [13, 2, 14, 1, 36, 55, 32, 25, 12]. These recordings provide insights that cannot be obtained by pooling multiple single-electrode recordings [27]. This is a significant development as it enables systematic investigations of large circuits of neurons to better understand their function and structure, as well as how they are affected by injury, disease, and pharmacological interventions [20].

On dense MEAs, each recording channel may record spikes from multiple, nearby neurons, while each neuron may leave an extracellular footprint on multiple channels. Inferring the spiking activity of individual neurons, a task called *spike sorting*, is therefore a challenging blind source separation problem, complicated by the large volume of recorded data [46]. Despite the challenges presented by

spike sorting large-scale recordings, its importance cannot be overstated as it has been shown that isolating the activity of individual neurons is essential to understanding brain function [35]. Recent efforts have concentrated on providing scalable spike sorting algorithms for large scale MEAs and already several methods can be used for recordings taken from hundreds to thousands of channels [42, 31, 10, 54, 22, 26]. However, scalability, and in particular automation, of spike sorting pipelines remains challenging [8].

One strategy for spike sorting on dense MEAs is to spatially localize detected spikes before clustering. In theory, spikes from the same neuron should be localized to the same region of the recording area (near the cell body of the firing neuron), providing discriminatory, low-dimensional features for each spike that can be utilized with efficient density-based clustering algorithms to sort large data sets with tens of millions of detected spikes [22, 26]. These location estimates, while useful for spike sorting, can also be exploited in downstream analyses, for instance to register recorded neurons with anatomical information or to identify the same units from trial to trial [9, 22, 41].

Despite the potential benefits of localization, preexisting methods have a number of limitations. First, most methods are designed for low-channel count recording devices, making them difficult to use with dense MEAs [9, 51, 3, 30, 29, 34, 33, 50]. Second, current methods for dense MEAs utilize cleaned extracellular action potentials (through spike-triggered averaging), disallowing their use before spike sorting [48, 6]. Third, all current model-based methods, to our knowledge, are non-Bayesian, relying primarily on numerical optimization methods to infer the underlying parameters. Given these current limitations, the only localization methods used consistently before spike sorting are simple heuristics such as a center of mass calculation [38, 44, 22, 26].

In this paper, we present a scalable Bayesian modelling approach for spike localization on dense MEAs (less than $\sim 50\mu$m between channels) that can be performed *before* spike sorting. Our method consists of a generative model, a data augmentation scheme, and an amortized variational inference method implemented with a variational autoencoder (VAE) [11, 28, 47]. Amortized variational inference has been used in neuroscience for applications such as predicting action potentials from calcium imaging data [52] and recovering latent dynamics from single-trial neural spiking data [43], however, to our knowledge, it has not been used in applications to extracellular recordings.

After training, our method allows for localization of one million spikes (from high-density MEAs) in approximately 37 seconds on a TITAN X GPU, enabling real-time analysis of massive extracellular datasets. To evaluate our method, we use biophysically realistic simulated data, demonstrating that our localization performance is significantly better than the center of mass baseline and can lead to higher-accuracy spike sorting results across multiple probe geometries and noise levels. We also show that our trained VAE can generalize to recordings on which it was not trained. To demonstrate the applicability of our method to real data, we assess our method qualitatively on real extracellular datasets from a Neuropixels [25] probe and from a BioCam4096 recording platform.

To clarify, our contribution is not full spike sorting solution. Although we envision that our method can be used to improve spike sorting algorithms that currently rely center of mass location estimates, interfacing with and evaluating these algorithms was beyond the scope of our paper.

## 2    Background

### 2.1    Spike localization

We start with introducing relevant notation. First, we define the identities and positions of neurons and channels. Let $\boldsymbol{n} := \{n_i\}_{i=1}^{M}$, be the set of $M$ neurons in the recording and $\boldsymbol{c} := \{c_j\}_{j=1}^{N}$, the set of $N$ channels on the MEA. The position of a neuron, $n_i$, can be defined as $p_{n_i} := (x_{n_i} y_{n_i}, z_{n_i}) \in \mathbb{R}^3$ and similarly the position of a channel, $c_j$, $p_{c_j} := (x_{c_j}, y_{c_j}, z_{c_j}) \in \mathbb{R}^3$. We further denote $p_{\boldsymbol{c}} := \{p_{c_j}\}_{j=1}^{N}$ to be the position of all $N$ channels on the MEA. In our treatment of this problem, the neuron and channel positions are single points that represent the centers of the somas and the centers of the channels, respectively. These positions are relative to the origin, which we set to be the center of the MEA. For the neuron, $n_i$, let $\boldsymbol{s}_i := \{s_{i,k}\}_{k=1}^{K_i}$, be the set of spikes detected during the recording where $K_i$ is the total number of spikes fired by $n_i$. The recorded extracellular waveform of $s_{i,k}$ on a channel, $c_j$, can then be defined as $w_{i,k,j} := \{r_{i,k,j}^{(0)}, r_{i,k,j}^{(1)}, ..., r_{i,k,j}^{(t)}, ..., r_{i,k,j}^{(T)}\}$ where $r_{i,k,j}^{(t)} \in \mathbb{R}$ and $t = 0, \ldots, T$. The set of waveforms recorded by each of the $N$ channels of the MEA during the

spike, $s_{i,k}$, is defined as $\boldsymbol{w}_{i,k} := \{w_{i,k,j}\}_{j=1}^N$. Finally, for the spike, $s_{i,k}$, the point source location can be defined as $p_{s_{i,k}} := (x_{s_{i,k}}, y_{s_{i,k}}, z_{s_{i,k}}) \in \mathbb{R}^3$.

The problem we attempt to solve can now be stated as follows: *Localizing a spike, $s_{i,k}$, is the task of finding the corresponding point source location, $p_{s_{i,k}}$, given the observed waveforms $\boldsymbol{w}_{i,k}$ and the channel positions, $p_{\boldsymbol{c}}$.*

We make the assumption that the point source location, $p_{s_{i,k}}$ is actually the location of the firing neuron's soma, $p_{n_i}$. Given the complex morphological structure of many neurons, this assumption may not always be correct, but it provides a simple way to assess localization performance and evaluate future models.

## 2.2 Center of mass

Many modern spike sorting algorithms localize spikes on MEAs using the center of mass or barycenter method [44, 22, 26]. We summarize the traditional steps for localizing a spike, $s_{i,k}$ using this method. First, let us define $\alpha_{i,k,j} := \min_t w_{i,k,j}$ to be the negative amplitude peak of the waveform, $w_{i,k,j}$, generated by $s_{i,k}$ and recorded on channel, $c_j$. We consider the negative peak amplitude as a matter of convention since spikes are defined as inward currents. Then, let $\boldsymbol{\alpha}_{i,k} := (\alpha_{i,k,j})_{j=1}^N$ be the vector of all amplitudes generated by $s_{i,k}$ and recorded by all $N$ channels on the MEA.

To find the center of mass of a spike, $s_{i,k}$, the first step is to determine the central channel for the calculation. This central channel is set to be the channel which records the minimum amplitude for the spike, $c_{j_{min}} := c_{\mathrm{argmin}_j \alpha_{i,k,j}}$ The second and final step is to take the $L$ closest channels to

$c_{j_{min}}$ and compute, $\hat{x}_{s_{i,k}} = \dfrac{\sum_{j=1}^{L+1}(x_{c_j})|\alpha_{i,k,j}|}{\sum_{j=1}^{L+1}|\alpha_{i,k,j}|}, \hat{y}_{s_{i,k}} = \dfrac{\sum_{j=1}^{L+1}(y_{c_j})|\alpha_{i,k,j}|}{\sum_{j=1}^{L+1}|\alpha_{i,k,j}|}$ where all of the $L+1$

channels' positions and recorded amplitudes contribute to the center of mass calculation.

The center of mass method is inexpensive to compute and has been shown to give informative location estimates for spikes in both real and synthetic data [44, 37, 22, 26]. Center of mass, however, suffers from two main drawbacks: First, since the chosen channels will form a convex hull, the center of mass location estimates must lie *inside* the channels' locations, negatively impacting location estimates for neurons outside of the MEA. Second, center of mass is biased towards the chosen central channel, potentially leading to artificial separation of location estimates for spikes from the same neuron [44].

# 3 Method

In this section, we introduce our scalable, model-based approach to spike localization. We describe the generative model, the data augmentation procedure, and the inference methods.

## 3.1 Model

Our model uses the recorded amplitudes on each channel to determine the most likely source location of $s_{i,k}$. We assume that the peak signal from a spike decays exponentially with the distance from the source, r: $a\exp(br)$ where $a, b \in \mathbb{R}, r \in \mathbb{R}^+$. This assumption is well-motivated by experimentally recorded extracellular potential decay in both a salamander and mouse retina [49, 22], as well as a cat cortex [16]. It has also been further corroborated using realistic biophysical simulations [18].

We utilize this exponential assumption to infer the source location of a spike, $s_{i,k}$, since localization is then equivalent to solving for $s_{i,k}$'s unknown parameters, $\theta_{s_{i,k}} := \{a_{i,k}, b_{i,k}, x_{s_{i,k}}, y_{s_{i,k}}, z_{s_{i,k}}\}$ given the observed amplitudes, $\boldsymbol{\alpha}_{i,k}$. To allow for localization without knowing the identity of the firing neuron, we assume that each spike has individual exponential decay parameters, $a_{i,k}, b_{i,k}$, and individual source locations, $p_{s_{i,k}}$. We find, however, that fixing $b_{i,k}$ for all spikes to a constant that is equal to an empirical estimate from literature (decay length of $\sim 28\mu m$) works best across multiple probe geometries and noise levels, so we did not infer the value for $b_{i,k}$ in our final method. We will refer to the fixed decay rate as $b$ and exclude it from the unknown parameters moving forward.

The generative process of our exponential model is as follows,

$$a_{i,k} \sim N(\mu_{a_{i,k}}, \sigma_a), \ x_{s_{i,k}} \sim N(\mu_{x_{s_{i,k}}}, \sigma_x), \ y_{s_{i,k}} \sim N(\mu_{y_{s_{i,k}}}, \sigma_y), \ z_{s_{i,k}} \sim N(\mu_{z_{s_{i,k}}}, \sigma_z)$$

$$\hat{\boldsymbol{r}}_{i,k} = \|(x_{s_{i,k}}, y_{s_{i,k}}, z_{s_{i,k}}) - p_{\boldsymbol{c}}\|_2, \ \boldsymbol{\alpha}_{i,k} \sim \mathcal{N}(a_{i,k} \exp(b\hat{\boldsymbol{r}}_{i,k}), I) \tag{1}$$

In our observation model, the amplitudes are drawn from an isotropic Gaussian distribution with a variance of one. We chose this Gaussian observation model for computational simplicity and since it is convenient to work with when using VAEs. We discuss the limitations of our modeling assumptions in Section 5 and propose several extensions for future works.

For our prior distributions, we were careful to set sensible parameter values. We found that inference, especially for a spike detected near the edge of the MEA, is sensitive to the mean of the prior distribution of $a_{i,k}$, therefore, we set $\mu_{a_{i,k}} = \lambda \alpha_{i,k,j_{min}}$ where $\alpha_{i,k,j_{min}}$ is the smallest negative amplitude peak of $s_{i,k}$. We choose this heuristic because the absolute value of $\alpha_{i,k,j_{min}}$ will always be smaller than the absolute value of the amplitude of the spike at the source location, due to potential decay. Therefore, scaling $\alpha_{i,k,j_{min}}$ by $\lambda$ gives a sensible value for $\mu_{a_{i,k}}$. We empirically choose $\lambda = 2$ for the final method after performing a grid search over $\lambda = \{1, 2, 3\}$. The parameter, $\sigma_a$, does not have a large affect on the inferred location so we set it to be approximately the standard deviation of the $\boldsymbol{\alpha}_{i,k,j_{min}}$ (50). The location prior means, $\mu_{x_{s_{i,k}}}, \mu_{y_{s_{i,k}}}, \mu_{z_{s_{i,k}}}$, are set to the location of the minimum amplitude channel, $p_{c_{j_{min}}}$, for the given spike. The location prior standard deviations, $\sigma_x, \sigma_y, \sigma_z$, are set to large constant values to flatten out the distributions since we do not want the location estimate to be overly biased towards $p_{c_{j_{min}}}$.

## 3.2 Data Augmentation

For localization to work well, the input channels should be centered around the peak spike, which is hard for spikes near the edges (edge spikes). To address this issue, we employ a two-step data augmentation. First, inputs for edge spikes are padded such that the channel with the largest amplitude is at the center of the inputs. Second, all channels are augmented with an indicating variable which provides signal to distinguish them for the inference network. To be more specific, we introduce virtual channels outside of the MEA which have the same layout as the real, recording channels (see appendix C). We refer to a virtual channel as an "unobserved" channel, $c_{j_u}$, and to a real channel on the MEA as an "observed" channel, $c_{j_o}$. We define the amplitude on an unobserved channel, $\alpha_{i,k,j_u}$, to be zero since unobserved channels do not actually record any signals. We let the amplitude for an observed channel, $\alpha_{i,k,j_o}$, be equal to $\min_t w_{i,k,j_o}$, as before.

Before defining the augmented dataset, we must first introduce an indicator function, $1_o : \alpha \rightarrow \{0, 1\}$:

$$1_o(\alpha) = \begin{cases} 1, & \text{if } \alpha \text{ is from an observed channel,} \\ 0, & \text{if } \alpha \text{ is from an unobserved channel.} \end{cases}$$

where $\alpha$ is an amplitude from any channel, observed or unobserved.

To construct the augmented dataset for a spike, $s_{i,k}$, we take the set of $L$ channels that lie within a bounding box of width $W$ centered on the *observed* channel with the minimum recorded amplitude, $c_{j_{o_{min}}}$. We define our newly augmented observed data for $s_{i,k}$ as,

$$\boldsymbol{\beta}_{i,k} := \{(\alpha_{i,k,j}, 1_o(\alpha_{i,k,j}))\}_{j=1}^{L} \tag{2}$$

So, for a single spike, we construct a $L \times 2$ dimensional vector that contains amplitudes from $L$ channels and indices indicating whether the amplitudes came from observed or unobserved channels.

Since the prior location for each spike is at the center of the subset of channels used for the observed data, for edge spikes, the data augmentation *puts the prior closer to the edge* and is, therefore, more informative for localizing spikes near/off the edge of the array. Also, since edge spikes are typically seen on less channels, the data augmentation serves to ignore channels which are away from the spike, which would otherwise be used if the augmentation is not employed.

## 3.3 Inference

Now that we have defined the generative process and data augmentation procedure, we would like to compute the posterior distribution for the unknown parameters of a spike, $s_{i,k}$,

$$p(a_{i,k}, x_{s_{i,k}}, y_{s_{i,k}}, z_{s_{i,k}} | \boldsymbol{\beta}_{i,k}) \tag{3}$$

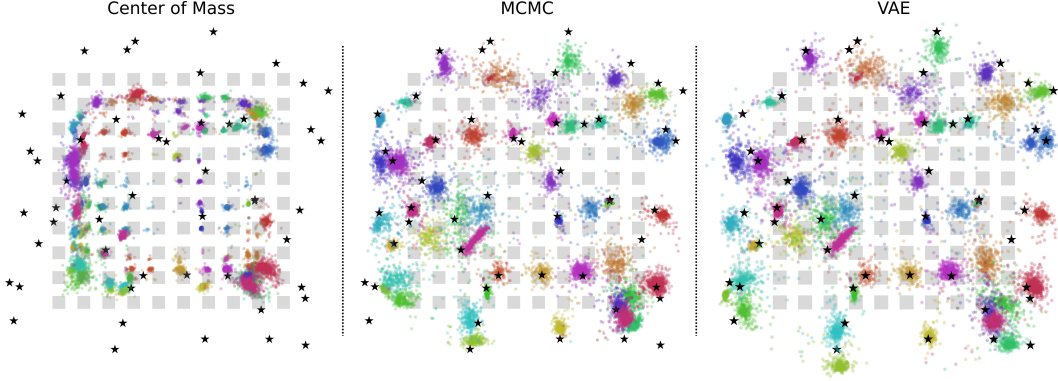

Figure 1: *Estimated spike locations for the different methods on a 10μV recording.* Center of mass estimates (left) are calculated using 16 observed amplitudes. The MCMC estimated locations (middle) used 9-25 observed amplitudes for inference, and the VAE model (right) was trained on 9-25 observed amplitudes and a 10 amplitude jitter (amplitude jitter is described in 3.3.3).

given the augmented dataset, $\boldsymbol{\beta}_{i,k}$. To infer the posterior distribution for each spike, we utilize two methods of Bayesian inference: MCMC sampling and amortized variational inference.

### 3.3.1 MCMC sampling

We use MCMC to assess the validity and applicability of our model to extracellular data. We implement our model in *Turing* [15], a probabilistic modeling language in Julia. We run Hamiltonian Monte Carlo (HMC) [39] for 10,000 iterations with a step size of 0.01 and a step number of 10. We use the posterior means of the location distributions as the estimated location.[1]

Despite the ease use of probabilistic programming and asymptotically guaranteed inference quality of MCMC methods, the scalability of MCMC methods to large-scale datasets is limited. This leads us to implement our model as a VAE and to perform amortized variational inference for our final method.

### 3.3.2 Amortized variational inference

To speed up inference of the spike parameters, we construct a VAE and use amortized variational inference to estimate posterior distributions for each spike. In variational inference, instead of sampling from the target intractable posterior distribution of interest, we construct a variational distribution that is tractable and minimize the Kullback–Leibler (KL) divergence between the variational posterior and the true posterior. Minimizing the KL divergence is equivalent to maximizing the evidence lower bound (ELBO) for the log marginal likelihood of the data. In VAEs, the parameters of the variational posterior are not optimized directly, but are, instead, computed by an inference network.

We define our variational posterior for $x, y, z$ as a multivariate Normal with diagonal covariance where the mean and diagonal of the covariance matrix are computed by an inference network

$$q_\Phi(x, y, z) = \mathcal{N}(\boldsymbol{\mu}_{\phi_1}(f_{\phi_0}(v_{i,k})), \boldsymbol{\sigma}^2_{\phi_2}(f_{\phi_0}(v_{i,k}))) \tag{4}$$

The inference network is implemented as a feed-forward, deep neural network parameterized by $\Phi = \{\phi_0, \phi_1, \phi_2\}$. As one can see, the variational parameters are a function of the input $\boldsymbol{v}$.

When using an inference network, the input can be any part of the dataset so for our method, we use, $\boldsymbol{v_{i,k}}$, as the input for each spike, $s_{i,k}$, which is defined as follows:

$$\boldsymbol{v}_{i,k} \coloneqq \{(w_{i,k,j}, 1_o(\alpha_{i,k,j}))\}_{j=1}^L \tag{5}$$

where $w_{i,k,j}$ is the waveform detected on the jth channel (defined in Section 2.1). Similar to our previous augmentation, the waveform for an unobserved channel is set to be all zeros. We choose to input the waveforms rather than the amplitudes because, empirically, it encourages the inferred location estimates for spikes from the same neuron to be better localized to the same region of the MEA. For both the real and simulated datasets, we used ∼2 ms of readings for each waveform.

| Method | Observed Channels | 2D Avg. Spike Distance from Soma ($\mu m$) | | |
|--------|-------------------|--------|--------|--------|
| | | 10 $\mu$V | 20 $\mu$V | 30 $\mu$V |
| COM | 4 | 15.84±10.08 | 16.46±10.39 | 17.18±10.97 |
| COM | 9 | 18.05±11.42 | 18.59±11.67 | 19.22±12.1 |
| COM | 16 | 20.94±13.09 | 21.54±13.46 | 22.17±13.94 |
| COM | 25 | 23.44±14.81 | 24.31±15.43 | 25.18±15.98 |
| MCMC | 9-25 | 9.87±8.64 | 11.30±9.85 | 13.31±11.67 |
| VAE - 0$\mu$V | 4-9 | 9.21±8.00 | 10.40±8.97 | 12.05±10.35 |
| VAE - 10$\mu$V | 4-9 | **8.79±7.49** | **9.79±8.31** | **11.18±9.56** |
| VAE - 0$\mu$V | 9-25 | 8.94±7.91 | 10.48±9.334 | 12.43±11.223 |
| VAE - 10$\mu$V | 9-25 | 9.12±7.83 | 10.41±9.07 | 12.27±10.78 |

Table 1: *Results for the 2D location estimates.* These results are for three simulated, square MEA datasets with noise levels ranging from $10\mu$V-$30\mu$V. For the VAE methods in the first column, the amount of amplitude jitter used is displayed to the right (amplitude jitter is described in 3.3.3).

The decoder for our method reconstructs the *amplitudes* from the observed data rather than the waveforms. Since we assume an exponential decay for the amplitudes, the decoder is a simple Gaussian likelihood function, where given the Euclidean distance vector $\hat{r}_{i,k}$, computed by samples from the variational posterior, the decoder reconstructs the mean value of the observed amplitudes with a fixed variance. The decoder is parameterized by the exponential parameters of the given spike, $s_{i,k}$, so it reconstructs the amplitudes of the augmented data, $\boldsymbol{\beta}_{i,k}^{(0)}$, with the following expression: $\hat{\boldsymbol{\beta}}_{i,k}^{(0)} := a_{i,k} \exp(b\hat{r}_{i,k}) \times \beta_{i,k}^{1}$ where $\hat{\boldsymbol{\beta}}_{i,k}^{(0)}$ is the reconstructed observed amplitudes. By multiplying the reconstructed amplitude vector by $\beta_{i,k}^{1}$ which consists of either zeros or ones (see Eq. 5), the unobserved channels will be reconstructed with amplitudes of zero and the observed channels will be reconstructed with the exponential function. For our VAE, instead of estimating the distribution of $a_{i,k}$, we directly optimize $a_{i,k}$ when maximizing the lower bound. We set the initial value of $a_{i,k}$ to the mean of the prior. Thus, $a_{i,k}$ can be read as a parameter of the decoder.

Given our inference network and decoder, the ELBO we maximize for each spike, $s_{i,k}$, is given by,

$$\log p(\boldsymbol{\beta}_{i,k}; a_{i,k}) \geq -\text{KL}\left[q_{\Phi}(x,y,z) \,\|\, p_x p_y p_z\right] + \mathbb{E}_{q_{\Phi}}\left[\sum_{l=1}^{L} \mathcal{N}(\beta_{i,k,l}^{0}|a_{i,k}\exp(b\hat{r}_{i,k}), I)\beta_{i,k,l}^{1}\right]$$

where KL is the KL-divergence. The location priors, $p_x, p_y, p_z$, are normally distributed as described in 3.1, with means of zero (the position of the maximum amplitude channel in the observed data) and variances of 80. For more information about the architecture and training, see Appendix F.

### 3.3.3 Stabilized Location Estimation

In this model, the channel on which the input is centered can bias the estimate of the spike location, in particular when amplitudes are small. To reduce this bias, we can create *multiple inputs* for the same spike where *each input is centered on a different channel*. During inference, we can average the inferred locations for each of these inputs, thus lowering the central channel bias. To this end, we introduce a hyperparameter, *amplitude jitter*, where for each spike, $s_{i,k}$, we create multiple inputs centered on channels with peak amplitudes within a small voltage of the maximum amplitude, $\alpha_{i,k,j}$. We use two values for the amplitude jitter in our experiments: $0\mu V$ and $10\mu V$. When amplitude jitter is set to $0\mu V$, no averaging is performed; when amplitude jitter is set to $10\mu V$, all channels that have peak amplitudes within $10\mu V$ of $\alpha_{i,k,j}$ are used as inputs to the VAE and averaged during inference.

## 4 Experiments

### 4.1 Datasets

We simulate biophysically realistic ground-truth extracellular recordings to test our model against a variety of real-life complexities. The simulations are generated using the `MEArec` [4] package which includes 13 layer 5 juvenile rat somatosensory cortex neuron models from the neocortical microcircuit collaboration portal [45]. We simulate three recordings with increasing noise levels

(ranging from $10\mu V$ to $30\mu V$) for two probe geometries, a 10x10 channel square MEA with a 15 $\mu$m inter-channel distance and 64 channels from a Neuropixels probe ($\sim$25-40 $\mu$m inter-channel distance). Our simulations contain 40 excitatory cells and 10 inhibitory cells with random morphological subtypes, randomly distributed and rotated in 3D space around the probe (with a 20 $\mu$m minimum distance between somas). Each dataset has about 20,000 spikes in total (60 second duration). For more details on the simulation and noise model, see Appendix G.

For the real datasets, we use public data from a Neuropixels probe [32] and from a mouse retina recorded with the BioCam4096 platform [24]. The two datasets have 6 million and 2.2 million spikes, respectively. Spike detection and sorting (with our location estimates) are done using the HerdingSpikes2 software [22].

## 4.2 Evaluation

Before evaluating the localization methods, we must detect the spikes from each neuron in the simulated recordings. To avoid biasing our results by our choice of detection algorithm, we assume perfect detection, extracting waveforms from channels near each spiking neuron. Once the waveforms are extracted from the recordings, we perform the data augmentation. For the square MEA we use $W = 20, 40$, which gives $L = 4\text{-}9, 9\text{-}25$ real channels in the observed data, respectively. For the simulated Neuropixels, we use $W = 35, 45$, which gives $L = 3\text{-}6, 8\text{-}14$ real channels in the observed data, respectively. Once we have the augmented dataset, we generate location estimates for all the datasets using each localization method. For straightforward comparison with center of mass, we only evaluate the 2D location estimates (in the plane of the recording device).

In the first evaluation, we assess the accuracy of each method by computing the Euclidean distance between the estimated spike locations and the associated firing neurons. We report the mean and standard deviation of the localization error for all spikes in each recording.

In the second evaluation, we cluster the location estimates of each method using Gaussian mixture models (GMMs). The GMMs are fit with spherical covariances ranging from 45 to 75 mixture components (with a step size of 5). We report the true positive rate and accuracy for each number of mixture components when matched back to ground truth. To be clear, our use of GMMs is not a proposed spike sorting method for real data (the number of clusters is never known apriori), but rather a systematic way to evaluate whether our location estimates are more discriminable features than those of center of mass.

In the third evaluation, we again use GMMs to cluster the location estimates, however, this time combined with two principal components from each spike. We report the true positive rate and accuracy for each number of mixture components as before. Combining location estimates and principal components explicitly, to create a new, low-dimensional feature set, is introduced in Hilgen (2017). In this work, the principal components are whitened and then scaled with a hyperparameter, $\alpha$. To remove any bias from choosing an $\alpha$ value in our evaluation, we conduct a grid search over $\alpha = \{4, 6, 8, 10\}$ and report the best metric scores for each method.

In the fourth evaluation, we assess the generalization performance of the method by training a VAE on an extracellular dataset and then trying to infer the spike locations in another dataset where the neuron locations are different, but all other aspects are kept the same ($10\mu$V noise level, square MEA). The localization and sorting performance is then compared to that of a VAE trained directly on the second dataset and to center of mass.

Taken together, the first evaluation demonstrates how useful each method is purely as a localization tool, the second evaluation demonstrates how useful the location estimates are for spike sorting immediately after localizing, the third evaluation demonstrates how much the performance can improve given extra waveform information, and the fourth evaluation demonstrates how our method can be used across similar datasets without retraining. For all of our sorting analysis, we use SpikeInterface version 0.9.1 [5].

## 4.3 Results

Table 1 reports the localization accuracy of the different localization methods for the square MEA with three different noise levels. Our model-based methods far outperform center of mass with any number of observed channels. As expected, introducing amplitude jitter helps lower the mean and

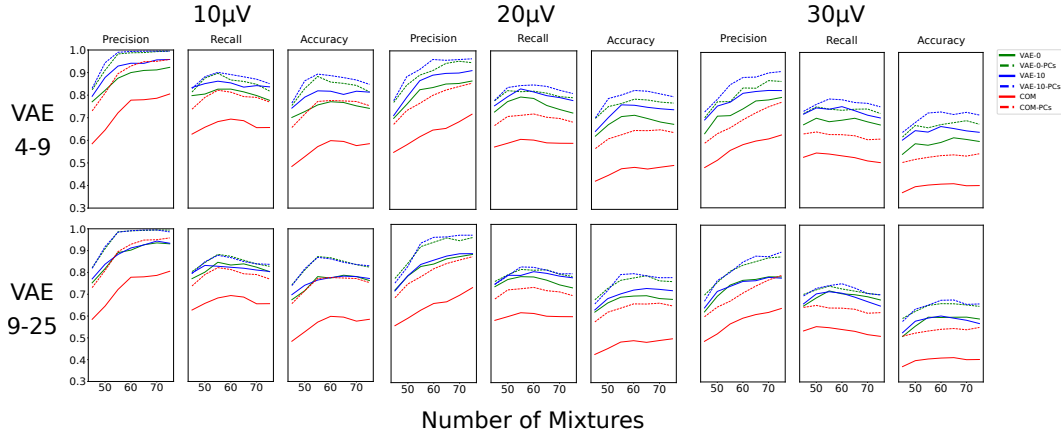

Figure 2: *Spike Sorting Performance on square MEA*. We compare the sorting performance of the VAE localization method and the COM localization method with and without principal components across all noise levels. For the VAE, we include the results with $0\mu$V and $10\mu$V amplitude jitter and with different amounts of observed channels (4-9 and 9-25). For COM, we plot the highest sorting performance (25 observed channels). The test data set has 50 neurons.

standard deviation of the location spike distance. Using a small width of $20\mu m$ when constructing the augmented data (4-9 observed channels) has the highest performance for the square MEA.

The location estimates for the square MEA are visualized in Figure 1. Recording channels are plotted as grey squares and the true soma locations are plotted as black stars. The estimated individual spike locations are colored according to their associated firing neuron identity. As can be seen in the plot, center of mass suffers both from artificial splitting of location estimates and poor performance on neurons outside the array, two areas in which the model-based approaches excel. The MCMC and VAE methods have very similar location estimates, highlighting the success of our variational inference in approximating the true posterior. See Appendix A for a location estimate plot when the VAE is trained and tested on simulated Neuropixels recordings.

In Figure 2, spike sorting performance on the square MEA is visualized for all localization methods (with and without waveform information). Here, we only show the sorting results for center of mass on 25 observed channels, where it performs at its best. Overall, the model-based approaches have significantly higher precision, recall, and accuracy than center of mass across all noise levels and all different numbers of mixtures. This illustrates how model-based location estimates provide a much more discriminatory feature set than the location estimates from the center of mass approaches. We also find that the addition of waveform information (in the form of principal components) improves spike sorting performance for all localization methods. See Appendix A for a spike sorting performance plot when the VAE is trained and tested on simulated Neuropixels recordings.

As shown in Appendix D, when our method is trained on one simulated recording, it can generalize well to another simulated recording with different neuron locations. The localization accuracy and sorting performance are only slightly lower than the VAE that is trained directly on the new recording. Our method also still outperforms center of mass on the new dataset even without training on it.

Figure 3 shows our localization method as applied to two real, large-scale extracellular datasets. In these plots, we color the location estimates based on their unit identity after spike sorting with HerdingSpikes2. These extracellular recordings do not have ground truth information as current, ground-truth recordings are limited to a few labeled neurons [56, 19, 21, 40, 54]. Therefore, to demonstrate that the units we find likely correspond to individual neurons, we visualize waveforms from a local grouping of sorted units on the Neuropixels probe. This analysis illustrates that are method can already be applied to large-scale, real extracellular recordings.

In Appendix E, we demonstrate that the inference time for the VAE is much faster than that of MCMC, highlighting the excellent scalability of our method. The inference speed of the VAE allows for localization of one million spikes in approximately 37 seconds on a TITAN X GPU, enabling real-time analysis of large-scale extracellular datasets.

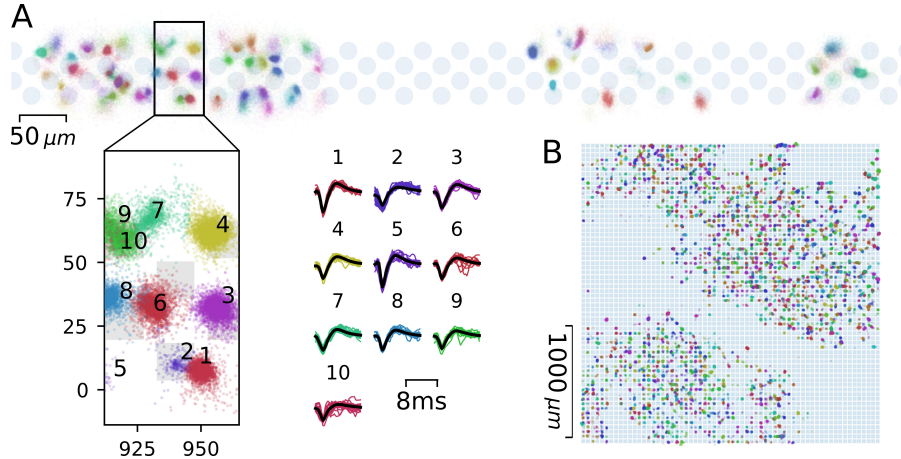

Figure 3: *Estimated spike locations for two real recordings.* A, Analysis of a one hour recording from an awake, head-fixed mouse with a Neuropixels probe. Spikes were detected using the HS2 package [22], their locations estimated using the VAE model, and clustered with mean shift, together with the first two principal components obtained from the waveforms. Shown are a large section of the probe, a magnification and corresponding spike waveforms from the clustered units. B, The same analysis performed on a recording from a mouse retina with a BioCam array from ref [24].

## 5 Discussion

Here, we introduce a Bayesian approach to spike localization using amortized variational inference. Our method significantly improves localization accuracy and spike sorting performance over the preexisting baseline while remaining scalable to the large volumes of data generated by MEAs. Scalability is particularly relevant for recordings from thousands of channels, where a single experiment may yield in the order of 100 million spikes.

We validate the accuracy of our model assumptions and inference scheme using biophysically realistic ground truth simulated recordings that capture much of the variability seen in real recordings. Despite the realism of our simulated recordings, there are some factors that we did not account for, including: bursting cells with event amplitude fluctuations, electrode drift, and realistic intrinsic variability of recorded spike waveforms. As these factors are difficult to model, future analysis of real recordings or advances in modeling software will help to understand possible limitations of the method.

Along with limitations of the simulated data, there are also limitations of our model. Although we assume a monopole current-source, every part of the neuronal membrane can produce action potentials [7]. This means that a more complicated model, such as a dipole current [50], line current-source [50], or modified ball-and-stick [48], might be a better fit to the data. Since these models have only ever been used *after* spike sorting, however, the extent at which they can improve localization performance *before* spike sorting is unclear and is something we would like to explore in future work. Also, our model utilizes a Gaussian observation model for the spike amplitudes. In real recordings, the true noise distribution is often non-Gaussian and is better approximated by pink noise models ($\frac{1}{f}$ noise) [53]. We plan to explore more realistic observation models in future works.

Since our method is Bayesian, we hope to better utilize the uncertainty of the location estimates in future works. Also, as our inference network is fully differentiable, we imagine that our method can be used as a submodule in a more complex, end-to-end method. Other work indicates there is scope for constructing more complicated models to perform event detection and classification [31], and to distinguish between different morphological neuron types based on their activity footprint on the array [6]. Our work is thus a first step towards using amortized variational inference methods for the unsupervised analysis of complex electrophysiological recordings.

## Footnotes

[1]The code for our MCMC implementation is provided in Appendix H.

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
