[Supplementary Material · DecayModelNeurIPSAppendix2019.pdf]

# Appendix

## A  Neuropixels Results

| Method | Observed Channels | 2D Avg. Spike Distance from Soma ($\mu m$) | | |
|---|---|---|---|---|
| | | 10 $\mu$V | 20 $\mu$V | 30 $\mu$V |
| COM | 4 | 23.85±12.95 | 25.16±14.21 | 26.66±15.6 |
| COM | 7 | 22.81±14.04 | 24.36±15.25 | 26.11±16.63 |
| COM | 12 | 26.33±15.55 | 28.26±16.66 | 30.1±17.81 |
| COM | 14 | 27.83±16.26 | 30.08±17.48 | 32.03±18.57 |
| MCMC | 8-14 | 14.28± 12.68 | 16.80±15.45 | 19.74±18.30 |
| VAE - 0$\mu$V | 3-6 | 14.25±12.88 | 15.74±14.88 | 18.44±17.68 |
| VAE - 10$\mu$V | 3-6 | 13.10±11.04 | **15.20±13.66** | **17.68±16.38** |
| VAE - 0$\mu$V | 8-14 | 13.31±12.46 | 15.63±15.51 | 18.49±18.89 |
| VAE - 10$\mu$V | 8-14 | **12.91±11.41** | 15.38±14.35 | 18.14±17.55 |

Table 2: *Results for the 2D location estimates.* These results are for three simulated, Neuropixels datasets with noise levels ranging from $10\mu$V-$30\mu$V. For the VAE methods in the first column, the amount of amplitude jitter used is displayed to the right (amplitude jitter is described in 3.3.2).

Figure 4: *Estimated spike locations for the different methods on a $10\mu V$ recording.* Center of mass estimates (top) are calculated using 7 channels. The MCMC estimated locations (middle) used 8-14 channels of observed amplitudes for inference, and the VAE model (bottom) was trained on 8-14 channels surrounding each spike and 0 amplitude jitter (see 3.3.2 for amplitude jitter explanation).

Figure 5: *Spike Sorting Performance on Neuropixels.* We compare the sorting performance of all localization methods with and without principal components across all noise levels. For the VAE, we include the results with and without amplitude jitter and with different amounts of real channels. For COM, we plot the highest sorting performance which was 4 observed channels.

## B  Effect of Noise on VAE

Figure 6: *Effect of noise on location inference for the VAE on the Neuropixels probe.* We vary the noise levels for the recording from $10\mu$V, $20\mu$V, and $30\mu$V. Increasing the noise also increases the number of outliers in and the spread of the location estimates.

Figure 7: *Effect of noise on location inference for the VAE on the square MEA.* We vary the noise levels for the recording from $10\mu$V, $20\mu$V, and $30\mu$V. Increasing the noise also increases the number of outliers in and the spread of the location estimates.

## C  Data Augmentation

Figure 8: *The simulated recording set-up and example data.* A, Example electrical traces from the MEA with recorded action potentials (spikes, negative deflections). B, The 2D layout of the simulated recording. Recording channels are indicated in grey, and the true locations of the simulated neurons in red. The traces in part A are taken from the first column of the array. Note each spike is visible in multiple channels, with a characteristic spatial decay. C, Illustration of the data augmentation procedure in cases where the spikes are detected on channels near the array boundary. A set of virtual channels is introduced, which are incapable of recording any signal, but would report non-zero amplitudes if they were present on the MEA.

# D   Generalization Performance

Table 3: Location results for the generalization performance of a VAE trained on one $10\mu$V, square MEA dataset and tested on another $10\mu$V, square MEA dataset. We compare the results of this VAE to another VAE that is trained directly on the second dataset to quantify the drop in performance when generalizing between datasets. We also compare to the center of mass baselines.

| Method | Observed Channels | 2D Avg. Spike Distance from Soma (microns) |
|---|---|---|
| COM | 4 | $16.53 \pm 10.83$ |
| COM | 9 | $18.25 \pm 13.0$ |
| COM | 16 | $20.41 \pm 14.57$ |
| COM | 25 | $22.73 \pm 16.32$ |
| VAE - 0 - Trained | 9-25 | $11.57 \pm 9.88$ |
| VAE - 0 - Inferred | 9-25 | $13.73 \pm 8.01$ |

Figure 9: *Spike Sorting Performance Generalization.* We compare the sorting performance of the VAE localization method and the COM localization method with and without principal components across all noise levels. For the VAE, we include the results with $0\mu$V and $10\mu$V amplitude jitter and with different amounts of observed channels (4-9 and 9-25). For COM, we plot the highest sorting performance (25 observed channels). The test data set has 50 neurons.

## E  Inference Time

Table 4: Results for the inference time of the VAE versus HMC sampling on the dataset. We ran HMC for 10,000 iterations. The VAE was run on a TITAN X GPU.

| Method | Per Spike Inference Time (s) | Dataset Inference Time (s) |
|--------|------------------------------|----------------------------|
| MCMC   | 0.343                        | 6669.0                     |
| VAE    | 0.000037                     | 0.722                      |

## F  Architecture and Training Details

We set the inference network to be 2 layers deep with ReLU nonlinearities. The hidden unit sizes in the inference network are set to be [500, 250]. We include batchnorm layers throughout the encoder to improve training and generalization.

We train the VAE with three different learning rates, $\{.0003, .001, .003\}$, and choose the learning rate that has the highest performance, although this parameter did not have a large effect on the results.

To ensure convergence for the simulated data, we train the network for 400 epochs on the entire dataset. For the real datasets, we train the network on a subset of the detected spikes ($\sim$100,000 spikes) and then we infer the rest of the locations.

## G  Simulated Data

To generate the extracellular recordings, we simulate the multi-compartment neuron models using NEURON [23] and use the transmembrane currents to compute extracellular action potentials (EAP) with LFPy [17]. EAPs are then combined with randomly generated spike trains to generate recordings. Finally, noise is added and the entire recording is filtered using a 3rd order Butterworth filter (0.3, 6 kHz).

For the noise model, we simulate templates for 300 neurons that are far away from the recording area. These small action potentials make up the background noise of the recording and have noise levels ranging from $10\mu V$ to $30\mu V$ standard deviation for the simulated datasets. We choose this noise model because it best captures the frequency and challenges of background noise in real extracellular recordings.

For each of the three recordings on one probe geometry, we fix the neuron locations to assess the effect of noise on the location estimates for each neuron.

## H  MCMC Turing Code

Below is the probabilistic program and inference code for the MCMC version of our method in Turing [15].

```julia
1   using Turing
2
3   # Define model
4   @model BayesianExpSpike(x_0, y_0, z_0, p_mean, p_std, locs, amps) = begin
5       S_a = 1
6       n_x ~ Normal(x_0, 80)
7       n_y ~ Normal(y_0, 80)
8       n_z ~ Normal(z_0, 80)
9
10      a ~ Normal(p_mean[1], p_std[1])
11      b = 0.035
12
13      r = sqrt.(sum((abs.(locs .- [n_x; n_y; n_z;])).^2; dims=1))
14      amps ~ MvNormal(vec(-a .* exp.(-b .* r)), S_a^0.5)
15  end
16
17  # Load data
```

```
18  real_channel_locs = ...
19  real_amps = ...
20  min_amp = ...
21  p_m = [2 * abs(min_amp)]
22  p_s = [50]
23
24  # Feed data into model
25  model = BayesianExpSpike(0, 0, 0, p_m, p_s, real_channel_locs, real_amps)
26
27  # Sampling
28  chn = sample(model, HMC(0.01, 10), 10_000)
```