[Reviews · NeurIPS 2019]

Reviewer 1



The paper is fairly clear and proposes a novel biologically inspired model for spike localization. Largely, it is well-down, and provides new paths for exploring the link between individual neurons and electrophysiological properties. It could be used later on for identifying properties of subtypes of neurons and their biological role, for instance, by matching multiple sensing techniques. However, there are a few issues. 1. To me, it's unclear why the data augmentation is truly necessary. Under the model, I feel like it would work without this step. An ablation analysis of what it actually accomplishes and a clear, precise description of why it is helpful would be beneficial. 2. The spike sorting analysis is frustrating for a number of reasons. First, the authors sweep over the number of clusters to report the results. The fact that the number of clusters is unknown a priori is one of the biggest issues, so this is unrealistic. Second, there is a complete lack of comparisons to state-of-the-art methods. Many, many methods are available with publicly available code for such datasets, including more useful evaluation metrics (e.g., [1]). The claim that combining location estimates and waveform estimation was introduced in 2017 is somewhat tenuous; this is implicit in nearly every dense MEA sorting method. Because of this, it is unclear to this reader whether the approach would actually contribute to a state-of-the-art sorting package. 3. The scalability here is through time, but does not appear to directly address scalability in channels. Specifically, a neuropixels device is a good current dense MEA, but several research groups are building and evaluating devices with >10,000 electrodes. Since the current VAE takes all channels as inputs and the number of detection typically scales linearly with the number of channel, I estimate this method would be quadratic with the number of channels. While certainly not an issue for most devices today, it would be useful to comment on. [1] Barnett, Alex H., Jeremy F. Magland, and Leslie F. Greengard. "Validation of neural spike sorting algorithms without ground-truth information." Journal of neuroscience methods 264 (2016): 65-77. The author feedback was reasonable to address my criticisms, and I have revised my score appropriately.

Reviewer 2



The paper introduces a generative model for predicting the location of spikes from microelectrode array recordings. Inference is performed by reformulating the learning algorithm as a Variational Autoencoder. While the model is well defined it bothers me that the amplitudes are modelled as Gaussian random variables. Spikes are essentially defined as non-gaussian events on the potential. Even though spike detection is not addressed in this work, I would expect the amplitude of the spikes to be modeled as non-gaussian in order to learn the appropriate structure. The paper seems to deal with localisation in a much more sophisticated way than anything else that is currently available. However, the numerical comparisons do not provide sufficient comparison with other localisation approaches. The paper would improve considerably through a better comparison with other modern localization approaches. All in all a good paper but it needs more comparison with state of the art. After reviewing the rebuttal: The reply for non-gaussianity of spikes was not satisfactory. If the authors are going to use a variational inference approach you should try to work on models with the appropriate prior distributions. Modern computational frameworks allow for more modelling flexibility than what is exposed in this work. I think the original score is sufficient for this work.

Reviewer 3



The authors develop an unsupervised, probabilistic, and scalable approach for spike localization from MEA recordings, in contrast to previous approaches which either required supervision, did not scale to large datasets, or relied on a simple heuristic (e.g., COM). Though the proposed model is relatively straightforward and the authors use standard approximate inference techniques to learn the desired posterior, the application to this domain and empirical validation of the approach seem to be novel contributions. The work is technically sound, with empirical results that demonstrate improved performance as compared to the COM heuristic for spike localization. However, there is no experimental validation of the proposed data augmentation scheme alone, as it seems that this scheme is used in both the MCMC and VAE approaches which makes it unclear what fraction of the performance improvement over COM is due to data augmentation versus the model-based posterior inference. A comparison to alternative spike localization methods besides COM would also strengthen the work, though I'm not sure if the works cited by the authors would even be feasible for the scale of the datasets being analyzed and therefore don't consider this a major shortcoming of the work. Though the writing is clear overall, some minor details could be clarified: the description of the model in Section 3.1 describes a procedure for choosing the location prior means (lines 122-123), but the proposed inference methods are stated as using a location prior mean of zero, which seems to be a discrepancy. Section 3.2 describes a bounding box of width W and number of channels L that are used in the data augmentation scheme, but the values for these used in the experiments are not explicitly stated (based on the captions from the figures/tables, it seems like values of W = 3, 5 and therefore L = 4-9, 9-25 were used, but this could be more made more clear). However, these are more minor issues that could easily be fixed. Update based on author feedback: Having read the authors' response, I feel like my concern about the effect of the data augmentation scheme was properly addressed, particularly if the authors commit to including an empirical analysis of the effect of the data augmentation on overall performance in the appendix as they mention. However, the other reviewers' comments on the lack of comparison to state-of-the-art methods makes me feel that this is more of a shortcoming of the submission than I had initially thought, and I'm not familiar enough with the alternative methods to know if leaving out any comparison to them is justified. Overall, I would still lean more towards accepting the submission but don't feel confident enough to strongly recommend acceptance, and therefore maintain my original score.

[Author Response · NeurIPS 2019]

Thank you very much for all for the thorough and thoughtful reviews. We want to address some of the concerns about our work and hopefully clear up any potential confusion as well.

**Reviewer 2** –

**Comparison to other algorithms** – One of the major complaints reviewer 2 had about our paper was the lack of comparison to other localization algorithms beyond center of mass (COM). To clarify, the task we were aiming to perform was *unsupervised*, *large-scale*, *pre-sorting* localization. Although we did cite other localization algorithms, these preexisting methods require spike sorting to be done before localization (to get average waveforms) and are also computationally expensive. For this reason, we wanted to compare to the only other localization method that is unsupervised, used consistently pre-sorting, and can scale to arbitrarily large datasets: COM. Our opinion is shared with reviewer 3 who also mentioned in his/her review that other localization algorithms would not scale to the large-scale datasets we were targeting. Hopefully this clarifies why we did not benchmark against other algorithms and alleviates reviewer 2's concern (we can include this reasoning about our benchmarks more clearly in the camera-ready version).

**Non-Gaussian amplitudes** – Reviewer 2 also mentioned he/she was concerned with our amplitudes being modelled as Gaussian random variables as spikes are non-Gaussian events. To clarify our modelling assumption, we chose a Gaussian amplitude model for convenience when working with a VAE and because the peak amplitude variations in the observed data seemed well-approximated by a Gaussian. It is a fair point that spikes are non-Gaussian, however, we were trying to model the peak amplitude variations rather than the actual spikes.

**Reviewer 1**-

**Scalability** – Reviewer 1 mentioned that our algorithm may not scale up to larger probes (with 10000+ channels) since we use all the channels as inputs to our model. However, our method does not actually use all the channels on the MEA. The observed data for our model is from a *subset* of channels that is centered on the channel with the largest negative amplitude. We construct this subset by taking a small radius (a hyperparameter that is typically set to 40-50 microns) of channels around the central channel. This subset consists of anywhere from 4 to 25 channels for the radii we chose in the paper. That means that our method would scale well to 10000+ channel MEAs.

**Data Augmentation** – Reviewer 1 mentioned that the data augmentation we implement seems unnecessary given our model. To clarify, the data augmentation only affects spikes detected near the edge of the array (spikes detected near the center of the array are unaffected by the augmentation) and provides two important benefits. First, since the prior location for each spike is at the center of the subset of channels used for the observed data, for spikes detected near the edge of the array, *the data augmentation actually puts the prior closer to the edge* and is, therefore, much more informative for localizing spikes near/off the edge of the array. Second, since spikes detected near the edge of the array typically have smaller amplitudes and are seen on fewer channels, our augmentation reduces the number of noisy channels with little signal in these cases. We experienced a performance increase with this data augmentation and would be happy to include an empirical analysis of this increase, if wanted, in the appendix of the camera-ready version.

**Spike Sorting** – Reviewer 1 is correct in saying that our "spike sorting" method of using a GMM and searching over the number of clusters would not be useable for real datasets where the number of clusters is unknown. However, since our work does not include the development of a new detection or classification algorithm (we solely focus on localization and location features) we just wanted to show that *our location features were more discriminable than those of COM*. This is why we show, for any number of mixtures in our GMMs, that our location features upper bound those of COM in both the true positive rate and accuracy of the clusterings. Therefore, any algorithm that uses centroid-based localization as features for their classification (Herding Spikes - Hilgen 2017, IronClust/JrClust - Jun 2017), could improve their accuracy by using our localization method instead. We actually use HerdingSpikes2 paired with our new localization method in our analysis of the real datasets to show how our method can be easily integrated into preexisting spike sorting pipelines. Since we did not develop new detection or clustering methods, we decided a GMM was the best and most direct way to show the advantages of improved localization, rather than comparing an integration into a full pipeline. For our claim that combining locations and waveforms was introduced in 2017, we were referring to how they were explicitly combined for clustering; reviewer 2 was correct about the implicit combination existing earlier.

**Reviewer 3**-

**Data Augmentation** – Reviewer 3 was interested in seeing the increase in performance over COM without the augmentation. As mentioned before, our augmentation only affects spikes detected near the edge and, despite that, we still see strong improvement over COM in the center of the MEA where the augmentation has no effect.

**Writing** - We can address the issues with experimental values and model discrepancies in the camera-ready version.

Thank you again to all the reviewers for their thoughtful comments and feedback. We believe our work provides a solid contribution to the electrophysiological community and can be both a tool and a benchmark for future localization work.

[Meta-Review · NeurIPS 2019]

Dear authors, congrats on the acceptance-- this paper was discussed extensively, and on balance was accepted. The reviewers provided multiple comments and it will be of critical importance that you revise the manuscript accordingly. One point that the reviewers felt was not deal with satisfactorily is the use of a Gaussian noise model for spike amplitudes, and I agree with that assessment-- at the very least this limitation should be transparently discussed, and work on how to use richer, non-Gaussian prior distributions should at least be cited. Finally, amortized inference and difference distributions on spike localizations/shapes have been studied in the related question of detecting action potentials from various imaging modalities (e.g. by Turaga and colleagues), and I do think it would be useful to link to that literature and the various methods that have been proposed in it.